# Investigation of the Properties of Polyphenylene Sulfone Blends

**DOI:** 10.3390/ma15186381

**Published:** 2022-09-14

**Authors:** Azamat Slonov, Ismel Musov, Azamat Zhansitov, Zhanna Kurdanova, Kamila Shakhmurzova, Svetlana Khashirova

**Affiliations:** Progressive Materials and Additive Technologies Center, Kabardino-Balkarian State University Named After H.M. Berbekov, St. Chernyshevsky, 173, 360004 Nalchik, Russia

**Keywords:** high-performance polymers, polyphenylene sulfone, polymer blends, rheological properties, mechanical properties, thermal properties

## Abstract

Polyphenylene sulfones (PPSU) blends with different viscosities have been studied. It is shown that the blends have a single-phase structure, regardless of the viscosities of the mixed polymers. It was found that blends having close values of the melt flow index (MFR) are also characterized by a similar melt viscosity in a wide range of shear rates, regardless of the viscosities of its constituent components. It has been found that PPSU blends with smaller MFR differences exhibit higher heat resistance and stability of mechanical properties, while blends with similar viscosity containing oligomers exhibit a broader molecular weight distribution (MWD) and have lower thermal and mechanical properties.

## 1. Introduction

Polysulfones are a group of heat-resistant polyarylenes that remain functional at temperatures from −100 to +250 °C [1]. Due to good mechanical and dielectric properties, as well as chemical and radiation resistance, polysulfones are widely used in the automotive industry, medicine, electrical engineering, and aircraft industry [2].

There are three main commercially available types of polysulfones: polysulfone (PSU), polyphenylene sulfone (PPSU), and polyethersulfone (PES). All of them contain aryl groups, which are linked by sulfo- (-SO2-) and ether groups (-O-) (Figure 1) [3].

PSU is mainly used for products requiring high rigidity, creep resistance, and good light transmission. PES has similar properties to PSU but is characterized by higher hardness and heat resistance. PPSU, compared to these materials, has a significantly higher impact strength and chemical resistance [4,5] and, in addition to traditional processing methods, is widely used in 3D printing [6,7].

PPSU has many applications in various industries: in medicine, it is used for sterilizable cases and trays, surgical instrument handles and other medical and dental devices; in plumbing for pressurized hot water supply systems; in the field of food service as fittings for beverage preparation, hot water boilers, window sights, and hoods; in the aircraft industry for aircraft cabin interiors [8].

Continuous technological developments lead to ever higher demands on materials and grades for various processing methods. The preparation of polymer blends is one of the simplest and cheapest ways to achieve the required physicochemical properties [9]. The vast majority of works devoted to PPSU blends refer to the development of membranes [10], which presents studies of blends of PPSU with PES [11], cellulose acetate (CA) [12], polyetherimide (PEI) [13], PSU [14] and polybenzimidazole (PBI) [15].

There is a rather limited number of works devoted to PPSU mixtures as engineering materials. In particular, the properties of blends of PPSU with polyphenylene sulfide (PPS) are studied in [16]. It was found that these polymers are partially compatible and that PPSU acts as an inhibitor of PPS crystallization, and therefore the elastic modulus of the mixtures is significantly reduced.

It was shown in [17] that a blend of PPSU and polyethylene terephthalate (PET) forms a two-phase structure, while at low concentrations of PET, mixtures have fairly good mechanical properties.

In [18], blends of PPSU with acrylonitrile butadiene styrene (ABS), high impact polystyrene (HIPS), and polystyrene (PS) were studied. It was found that the introduction of ABS and HIPS leads to a significant decrease in ductility, while with polycarbonate (PC) the impact strength and relative elongation increase, despite the thermodynamic incompatibility of these polymers.

An interesting direction is the study of compatible polymer-oligomer blends. The introduction of oligomers can lead to various physicochemical effects, such as broadening of the glass transition region [19], more complex rheological behavior at a large difference in the glass transition temperatures (Tg) of the components [20], and of particular interest is the plasticization of polymers with high viscosity. In particular, it was shown in [21] that the introduction of up to 10 wt. % PC oligomer in a commercial grade of high molecular weight PC leads to a significant increase in the melt flow rate (MFR) and the preservation of mechanical properties.

In [22], the influence of the PPSU oligomer on the main properties of high molecular weight PPSU and composites based on it was studied. It was found that the introduction of the oligomer leads to a significant increase in the MFR and a slight decrease in the Tg. At oligomer concentrations up to 20%, the mixtures demonstrate sufficiently high mechanical properties, however, a further increase in its content leads to a sharp decrease in impact strength and relative elongation.

Additionally, in [23], the PPSU oligomer was used for the plasticization of polyetherimide (PEI) and composites based on it. It was found that the PPSU oligomer is thermodynamically compatible with PEI and forms a single-phase structure. The introduction of an oligomer leads to a significant decrease in the viscosity of the PEI melt and an increase in the mechanical properties of composites based on it.

As a rule, the grade range of polymeric materials is based on the melt viscosity or its reciprocal value—MFR, which allows the consumer to purchase the most suitable grade for his processing method. According to the technological properties, polymer grades can be conditionally divided into the following groups: materials for processing by pressing (MFR value up to about 0.3 g/10 min), extrusion grades (MFR values from about 0.3 to 3 g/10 min), grades for injection molding (MFR from about 5 to 30 g/10 min) [24].

Kabardino-Balkarian State University named after H.M. Berbekov organized small-scale production of experimental batches of high-performance polymers, including polysulfones, which are synthesized with MFR from 0.5 to 500 g/10 min.

The well-established technique makes it possible to obtain polymers of the required viscosity with high repeatability [25], however, it is of interest to study the possibility of achieving a certain viscosity by mixing polymers with different MFRs. Such an approach will make it possible to control the rheological properties not only in the course of synthesis, but also in the presence of already synthesized materials, and will also open up the possibility of modifying industrial grades of polysulfones.

This method proved to be quite effective in the case of polyetheretherketone (PEEK): the possibility of controlling the rheological properties by mixing PEEK with different MFR was established, and it was found that blends with a similar melt viscosity are characterized by similar mechanical and thermal properties, regardless of the difference in the MFR of the mixed components [26].

Thus, the purpose of this study is to study the properties of mixtures of synthesized PPSUs with different MFRs.

## 2. Materials and Methods

Synthesized PPSUs with different MFRs were taken as objects of study. Synthesis of PPSU was carried out in a 500 mL three-necked flask equipped with a nitrogen inlet, a mechanical stirrer, a Dean–Stark trap, and a reflux condenser; 4,4′-dihydroxydiphenyl, 4,4′-dichlorodiphenyl sulfone, and potassium carbonate were charged in a flask. Then, N,N-dimethyl acetamide was added as a reaction solvent. The reaction mixture was gradually heated to 165 °C for 4 h to distill the water. After the temperature reached 165 °C, the reaction mixture was allowed to proceed at this temperature for 6 h. After synthesis, the mixture was discharged, and the formed salts were filtered. Then, the reaction solution was slowly poured into the water acidified with oxalic acid. The precipitated polymer was filtered and washed several times with water and dried in a vacuum oven at 160 °C for about 12 h. The viscosity of the polymers was controlled by varying amounts of excess 4,4′-dichlorodiphenyl sulfone

Composites were obtained by mixing in the melt on a PJSZ twin-screw microextruder from Ningbo Haitai Machinery Co., Ltd (Ningbo, China) with L/D = 30, at a maximum temperature of 360 °C. Test samples were obtained by injection molding on an SZS-20 injection molding machine from Ningbo Haitai Machinery Co., Ltd (Ningbo, China) at a material cylinder temperature of 380–390 °C and a mold temperature of 180 °C.

Mechanical tests for uniaxial tension were performed on dog-bone samples with dimensions according to GOST 112 62-80. The tests were carried out on a universal testing machine CT-TCS 2000 from Gotech Testing Machines Inc. (Taichung, Taiwan), at a temperature of 23 °C. Impact tests were performed with and without a notch, according to the Izod method according to GOST 19109-84 on the device GT-7045-MD from Gotech Testing Machines Inc. (Taichung, Taiwan), with pendulum energy of 11 J.

The melt flow index (MFR) was determined on a PTR-LAB-02 instrument manufactured by LOIP (St. Petersburg, Russia) at a temperature of 350 °C and a load of 5 kgf. The viscosity of the melt was determined on a capillary rheometer LCR 7001 from Dynisco (Franklin, VA, USA) at a temperature of 350 °C.

The glass transition temperature was determined by differential scanning calorimetry (DSC) on a DSC 4000 device from PerkinElmer (Waltham, MA, USA) in the air at a heating rate of 10 °C/min, as well as by dynamic mechanical analysis (DMA) at a heating rate of 2 °C/min and a frequency 1 Hz on a DMA-1 from Mettler Toledo (Zurich, Switzerland).

Vicat heat resistance was determined on a Gotech Testing Machine HV-3000-D3 instrument (Taichung Industry Park, Taichung City, Taiwan) according to GOST 15088-83.

Thermal stability was determined by thermogravimetric analysis (TGA) on TGA 4000 instrument from PerkinElmer (Waltham, MA, USA) in the air at a heating rate of 10 °C/min.

The synthesized PPSU with an MFR of 17.7 g/10 min was taken as a reference sample; further, by mixing polymers with different viscosities, an MFR close to the value of the reference sample was achieved. With an increase in the difference in the MFR of the mixed PPSU, the compositions were adjusted to provide the required MFR of the mixture in accordance with the task.

## 3. Results and Discussion

Table 1 shows that higher viscosity (or lower MFR) PPSU samples have higher Tg. These results are consistent with the known patterns in the relationship between the molecular weight (MW) and thermal properties of polymers [25,27]. The lower the molecular weight, the lower the Tg and the higher the MFR. These properties are very sensitive to changes in MW and can serve as parameters for estimating molecular weight characteristics.

As the difference in the MFR of the mixed polymers increases, the Tg of the mixtures decreases uniformly, which is apparently due to the introduction of more and more low molecular weight components despite the reduction in its amount. At the same time, the blends demonstrate thermodynamic compatibility, showing only one glass transition peak on the DSC curves (Figure 1).

It is noteworthy that the Tg values of the PPSU-6/38 and PPSU-2/54 mixtures are closer to the Tg values of their components with higher MFR. A further increase in the difference in the MFR of the mixed materials (a blend PPSU-3/292) leads to averaging of Tg, and when using 20% low molecular weight PPSU-n/f, the value of Tg of the blend (PPSU-3/n/f) sharply decreases to 201.8 °C, which is 8.6% lower than the Tg value of pure PPSU-17.

Figure 2 shows the dependence of the storage moduli (E’) of PPSU blends on temperature. E’ of a polymer material is proportional to the accumulated energy under mechanical action and is a characteristic of the material’s ability to withstand mechanical loads [28]. As can be seen from Figure 2, when a certain temperature range is reached, a sharp decrease in the storage modules occurs. The beginning of this effect can be taken as Tg.

Tg was also determined from the loss moduli (E″) of PPSU blends. The peak on the curve E″ was taken as the value of Tg. Figure 2 shows that with an increase in the difference in the MFR of the mixed components, the E″ peak shifts to lower temperatures.

Thus, from Figure 3 it can be seen that Tg determined by different methods has different values [29]. Since the glass transition phenomenon has a relaxation nature, Tg largely depends on the methods and conditions of determination, in particular, the heating rate and the frequency of mechanical action (in the case of DMA). In addition, the difference in values is due to the fact that the glass transition is a specific area and not a single temperature point. The DMA method is more sensitive to changes in the mobility of macromolecules, which immediately affects the mechanical properties, expressed in a change in the elastic modulus by several orders of magnitude, in contrast to the DSC method, where the change in heat capacity occurs by only 10–30%.

In general, regardless of the method used to determine Tg, there is an obvious tendency toward reduction as the difference in viscosities of mixed PPSU increases, which indicates an increase in the mobility of macromolecules in blends containing low molecular weight oligomers. At the same time, a decrease in heat resistance symbate with Tg is observed (Figure 3), which is natural for unfilled amorphous polymers.

It is known that MFR is the flow rate of a material at a fixed pressure during its flow through a capillary of a certain diameter at a certain temperature. However, the MFR value does not characterize the rheological behavior of the polymer material over the entire range of processing conditions but corresponds to only one point on the curve of melt viscosity versus shear rate [30]. Based on this, it was of interest to study the features of the flow of blends with a close MFR in a wide range of shear rates. These studies were carried out using a capillary rheometer, in a range of shear rates, covering various processing methods—from pressing to injection molding.

A very close rheological behavior of pure PPSU, blending with similar MFRs, was revealed, regardless of composition (Figure 4), which indicates that blends with close MFR values are also characterized by similar values of melt viscosity in a wide range of shear rates, and the introduction of oligomers leads to effective plasticization.

The study of thermal stability using the TGA method showed that PPSU blends have a fairly close temperature at the onset of degradation (Figure 5a). From Table 2 it can be seen that blends with oligomers (PPSU-3/292 and PPSU-3/n/f) have a slightly lower temperature loss of 5% mass. A more detailed study of thermal stability (Figure 5b) shows that there are two peaks on the derivative of mass loss curves, which indicates that the decomposition takes place in two stages. Temperatures of maximum decomposition rate (T_max_) lie in the range of approximately 500 to 630 °C and 630 to 750 °C. The first stage is due to chain breaking reactions and the loss of volatile components, the second stage is due to oxidation.

Table 2 shows that the temperatures of the maximum decomposition rate of PPSU blends at the first stage are quite close, however, in the second stage, blends with oligomers (PPSU-3/292 and PPSU-3/n/f) have a slightly lower temperature at the maximum decomposition rate, which indicates their insignificantly lower resistance to oxidation. In general, all materials have sufficiently high thermal stability.

The mechanical properties of PPSU blends were also studied (Table 3). Table 3 shows that blends PPSU-6/38 and PPSU-2/54 have high impact resistance without a notch, which is not inferior to pure PPSU-17, however, blends PPSU-3/292 and PPSU-3/n/f are characterized by lower values impact strength. Despite the fact that some samples of the PPSU-3/n/f blends do not break down when tested without notching, the lower stability of their properties and lower resistance to high-speed loads are obvious. Nevertheless, the blends have a fairly good impact resistance.

A study of notched impact strength shows that PPSU-6/38 and PPSU-2/54 blends have a significantly higher crack resistance compared to pure PPSU-17. Apparently, this is due to the high molecular weight polymers included in the blends and the small difference in the viscosities of the mixed components. The use of oligomers in blends of PPSU-3/292 and PPSU-3/n/f leads to a significant decrease in notched impact strength, despite their relatively low content. Thus, the introduction of brittle oligomers leads to a decrease in crack resistance, the more so, the lower the molecular weight of the oligomer used.

Table 3 also shows a slight increase in the elastic modulus of the blends with an increase in the difference in the MFR of the mixed PPSU. This fact, apparently, is also associated with the use of low molecular weight oligomers, since previous studies have established that the lower the molecular weight, the higher the elastic modulus of PPSU [31]. In addition, the strength is significantly increased: the tensile yield strength of the PPSU-3/n/f blend exceeds the corresponding value of pure PPSU-17 by 14.7%, and the tensile strength by about 9%. Notably, even when using a low molecular weight oligomer, the blend exhibits plastic flow despite a lower elongation at break (Figure 6).

A study was also made of the molecular weight distribution (MWD) of pure PPSU-17 and a blend containing the lowest molecular weight oligomer (PPSU-3/n/f). As can be seen, PPSU-17 is characterized by a unimodal MWD (Figure 7a). In the case of a mixture of PPSU-3/n/f, one can observe a distinct polymodal distribution (Figure 7b): there is a rather narrow peak corresponding to the high molecular weight fraction, which obviously refers to the component with high viscosity (PPSU-3) and two peaks corresponding to the low molecular weight fraction, which is the result of the introduction of the oligomer (PPSU-n/f). Apparently, the observed decrease in plastic and thermal properties is due to the presence of a large proportion of the low molecular weight fraction.

## 4. Conclusions

Thus, it can be concluded that achieving close values of MFR by mixing high and low molecular weight PPSU makes it possible to obtain materials with a similar melt viscosity in a wide range of shear rates. The use of oligomers leads to effective plasticization of high molecular weight PPSU, however, in this case, there is a slight decrease in impact strength without a notch and a significant drop in impact resistance with a notch, while at the same time, the strength and rigidity of the material increase. Additionally, the disadvantage of blends with oligomers is their lower glass transition temperature and, as a result, lower heat resistance.

In general, this method can be recommended for modifying PPSU with unsatisfactory rheological properties for processing by injection molding and 3D printing. Polymer blends with a small difference in MFR show higher dimensional and property stability. However, achieving the required rheological properties using low molecular weight oligomers is accompanied by a more noticeable decrease in thermal and mechanical characteristics.

Due to the combination of high mechanical and thermal properties, as well as the preservation of transparency, the blends with an optimal composition can be used to produce parts for various applications, including automotive construction, the food and household sectors, electrical engineering, and electronics, as well as in heating and sanitary engineering.

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
