# Peer review of "Investigation of the Properties of Polyphenylene Sulfone Blends"

_materials, 2022, doi:10.3390/ma15186381_

Round 1

Reviewer 1 Report

Dear Editor,

I received the manuscript “Investigation of the properties of polyphenylene sulfone blends” for review.

The authors presented promising and well-prepared research. I have only one suggestion. In tables and figures, the comma (in numbers) should be replaced by a dot. This is necessary to maintain a uniform format of the manuscript. In my opinion, this manuscript should be accepted for publication after minor revision.

I received the manuscript "Investigation of the properties of polyphenylene sulfone blends" for review. The manuscript reports on the synthesis and physical properties of polypropylene sulfones blends. The ability to control of the rheological properties by mixing of polypropylene sulfones with different values of the melt flow index (MFR) has also been shown. Synthesized polypropylene sulfones blends with smaller MFR differences demonstrate high heat resistance and stability of mechanical properties. I have only a few suggestions for improving this manuscript: - A description of the application of fabricated polypropylene sulfones should be included in Abstract and Introduction. - The experimental part should be completed with a more detailed synthesis, including a description of the separation and/or purification of polymers. - In tables and figures, the comma (in numbers) should be replaced by a dot. This is necessary to maintain a uniform format of the manuscript. In my opinion, this manuscript should be accepted for publication after minor revision.

Author Response

Dear Reviewer, thank you very much for your helpful comments. We kindly listened to your suggestions and made the necessary corrections to the article. In particular:

  • we added to the introduction information about the application of PPSU;
  • we supplemented the experimental part with more detailed information on the synthesis and purification of the polymer;
  • we replaced commas with dots in tables and figures.

Reviewer 2 Report

To easily express this study to the readers, the revisions below are suggested to be applied:

-        Author elaborated on the advantage and disadvantage of the material used, but applications for this study was not mentioned.  The authors do not elaborate on the importance of applying blends in which application. Mechanical properties, etc.??  

-        The English of the manuscript should be improved.

-        Please change the elastic modulus in Figure 2 as the storage modulus

-        Abbreviated names of the polymers in the introduction part should be written in parentheses—for example, polystyrene (PS).

-        Please add derivative mass loss for TGA analyses and discuss.

-        Results and discussion part of the manuscript include poor discussion. It should be improved.

-        Morphological analyses should also be carried out to evaluate the compatibilization of the samples. 

Author Response

Dear Reviewer, we thank you for your helpful and valuable comments. We took into account your comments and tried to improve the manuscript. In particular:

  • we supplemented the introduction and conclusions with information on the use of PPSU and possible applications of the obtained blends;
  • we corrected the title of the axis in fig. 2 as "modulus" because the figure shows both the storage modulus and the loss modulus;
  • we added abbreviated names of the polymers in the introduction part;
  • we added derivative mass loss and discussed them;
  • based on the results obtained and the results of previous studies, high-molecular polyphenylene sulfones and polyphenylene sulfone oligomers will form true solutions, since they have an identical chemical structure. Their thermodynamic compatibility makes it possible to obtain transparent samples of mixtures that have only 1 relaxation transition during thermal analysis. Based on this, change in morphology does not occur or is difficult to notice. In this regard, we have given in the article the most revealing types of analysis of properties.

Round 2

Reviewer 2 Report

The revised version of the manuscript is at a satisfactory level. Therefore, my decision is to accept this manuscript for possible publication in Materials